# A Comprehensive Evaluation of Microbial Synergistic Metabolic Mechanisms and Health Benefits in Kombucha Fermentation: A Review

**DOI:** 10.3390/biology14080952

**Published:** 2025-07-28

**Authors:** Xinyao Li, Norzin Tso, Shuaishuai Huang, Junwei Wang, Yonghong Zhou, Ruojin Liu

**Affiliations:** 1Key Laboratory of Biodiversity and Environment on the Qinghai-Tibetan Plateau, Ministry of Education, Xizang University, Lhasa 850000, China; 18254103778@163.com (X.L.); norzin0722@126.com (N.T.); s.huang1@utibet.edu.cn (S.H.); jwyx12240315@126.com (J.W.); 2School of Ecology and Environment, Xizang University, Lhasa 850000, China

**Keywords:** kombucha, fermentation, bioactive compounds, health effects, metabolic mechanisms

## Abstract

Kombucha, a traditional fermented tea, has gained attention for its potential health benefits, including antioxidant, antimicrobial, and immune-modulating effects. This review explores the complex fermentation process of kombucha, focusing on the interactions among microbial communities, such as *yeast*, *acetic acid bacteria*, and *lactic acid bacteria*, and their role in producing bioactive compounds. These compounds are essential for the health benefits attributed to kombucha. The review also examines how various fermentation conditions—such as temperature, time, and strain selection—affect microbial composition and the resulting bioactive components. Although kombucha’s health-promoting effects are widely recognized, the mechanisms behind these benefits remain inadequately understood. This review provides a comprehensive synthesis of the current research on kombucha fermentation, highlighting the importance of optimizing fermentation conditions to enhance its bioactive content and health effects. These findings contribute to the development of more effective fermentation processes and offer directions for future research, aiming to maximize the functional potential of kombucha as a health-promoting beverage.

## 1. Introduction

Kombucha, a traditional fermented beverage, originated in China and other East Asian regions and has spread widely across the globe over the centuries [1,2]. It is produced through the fermentation of tea leaves and sugar in solution, generating a variety of bioactive compounds with unique flavor profiles. These compounds are mainly produced by the combined action of microbial communities such as *Saccharomyces*, *Acetobacter*, and *Lactobacillus* [3]. Bioactive components, including ethanol, acetic acid, and organic acids, are closely related to the flavor characteristics and health effects of kombucha microbiota [4,5]. In addition, phenolic compounds and other active substances in kombucha contribute to its significant antioxidant, antimicrobial, and immune-modulating properties. These health benefits have made kombucha microbiota one of the key areas of global research interest [6,7,8].

Although the health benefits of kombucha microbiota have been widely recognized, their complex fermentation mechanisms and interactions among microbial communities remain to be explored. Existing studies have mostly focused on the exploration of single components or local metabolic pathways of kombucha microbiota, and fewer review articles have deeply and systematically sorted out the synergistic metabolic roles of microbial communities during the fermentation process of kombucha microbiota and the interrelationships with the fermentation conditions. Different fermentation conditions, such as fermentation time, temperature, and strain selection, can affect the composition of microbial communities, as well as the final product distribution, which, in turn, affects the bioactive components and quality characteristics of kombucha microbiota. However, existing studies still have not provided a deep enough understanding of how to optimize the health benefits by regulating the fermentation conditions.

Therefore, this review aims to integrate the existing research results, deeply explore the interactions of microbial communities during the fermentation process of kombucha microbiota, and analyze the regulation of fermentation conditions on the quality and bioactive components of kombucha microbiota. Through systematic literature sorting and analysis, this review provides an important theoretical basis for the optimization of the fermentation process and the development of functional beverages of kombucha microbiota, as well as new perspectives and directions for future related research.

## 2. Methods for Review and Word Frequency Analysis

To conduct a comprehensive and systematic analysis of the core research topics in the field of kombucha, this review is based on the manual reading of 32 relevant articles, from which keywords were extracted from each paper. The selected articles were sourced from multiple academic databases, including Google Scholar, ScienceDirect, PubMed, and others. All of them are peer-reviewed research papers, covering various aspects of kombucha, including its fermentation process, nutritional components, and health benefits.

During the keyword extraction process, three researchers independently reviewed the abstracts, methodologies, research content, and conclusions of each article. Based on the core points of the articles, they selected 30 to 50 keywords that best represented the research themes and key findings of each study. This approach ensured that the chosen keywords accurately reflected the research direction and contributions of the literature.

Subsequently, this study utilized Excel to perform a frequency analysis of the extracted keywords and created a word cloud to visualize the distribution of keyword frequencies. The analysis results revealed that the primary research hotspots in the field of kombucha focus on areas such as “fermentation”, “bioactive compounds”, “acetic acid”, and “antioxidant activity”, reflecting the core issues and trends within this field of study (Figure 1).

In addition, areas such as “fermentation process”, “*yeast*”, “*acetic acid bacteria*”, “*lactic acid bacteria*”, and “bioactive components” have also garnered significant attention. Therefore, this study focuses on analyzing and summarizing the fermentation and product processing of kombucha, the mechanisms of microbial interactions, and the health-promoting bioactive components.

The frequency analysis provided a solid foundation for the structured organization of this review and laid the theoretical groundwork for the subsequent discussion on the microbial interactions during the kombucha fermentation process and their impact on health benefits.

## 3. Fermentation Process and Product Processing

To prepare kombucha, the following ingredients are needed: tea leaves, white sugar, and pure water. The tea leaves and sugar should be weighed at a specific ratio and thoroughly mixed. The mixture is then placed in a sterile kettle to which pure water is added. The solution is brought to a boil and steeped to form the tea infusion. Afterward, the resulting tea-sugar solution is filtered and transferred into a sterile glass bottle. Once the solution has cooled to room temperature, the kombucha culture (SCOBY, symbiotic culture of bacteria and yeast) is added. The beverage is then incubated at an appropriate temperature until a biofilm forms. Finally, the fermented kombucha is filtered and separated to obtain both the liquid culture and the microbial film. Both of these products can be used in various fields for product processing. For instance, a liquid culture is applied in the food industry to produce health drinks, while a microbial film is utilized in industries such as textiles and food (Figure 2).

## 4. Fermentation Conditions of Kombucha

### 4.1. Source and Quality of the Strains

Kombucha strains from different sources exhibit significant differences in antioxidant capacity, which can be attributed to the specificity of the microbial community composition. This was further confirmed in free radical scavenging experiments [9]. The diversity and specificity of the microbial community structure influence the composition of metabolic products in the culture medium, which, in turn, affects the synthesis efficiency of antioxidant compounds. This ultimately results in notable differences in the antioxidant properties of kombucha strains from different sources.

### 4.2. Fermentation Time

The fermentation time of kombucha is correlated with the dynamic changes in its chemical composition, and it significantly affects the quality of the kombucha. Phenolic compounds, as the main antioxidant active components in kombucha, are directly responsible for the functional quality characteristics of the beverage. Relevant studies have shown that the total phenolic content increases linearly over the fermentation time, reaching a peak concentration on day 15. This may be related to the microbial-mediated biotransformation of macromolecular polyphenols, such as theaflavins, and the synthesis of small-molecule phenolic derivatives [9]. Additionally, research has found that the organic acid content increases with the fermentation process, while the pH gradually decreases [10]. However, prolonged fermentation, which leads to the pH exceeding the sensory threshold range, can cause flavor deterioration. Observations using the plate dilution method revealed that *Saccharomyces* proliferates rapidly at the beginning of fermentation, peaks, and then sharply declines. During this time, *Acetobacter* become active, resulting in an increase in organic acids and a subsequent decrease in pH. The length of fermentation alters the microbial community structure and the abundance of dominant microorganisms, which, in turn, affects both the fermentation process and the quality of the final product [11].

### 4.3. Fermentation and Storage Temperature

Temperature parameters during both the fermentation and storage processes of kombucha play a dual regulatory role in its quality. Fermentation temperature has a significant impact on the quality of kombucha, with 25 °C being identified as the optimal fermentation temperature [10]. Under this condition, the glucuronic acid content reaches its peak, and the sensory quality score is significantly higher compared to other temperature groups.

Additionally, the temperature during kombucha storage affects microbial growth, the stability of tea components, and antioxidant capacity [12]. Storage of kombucha microbiota includes inactivated and uninactivated storage. In the case of inactivation, the microorganisms in the kombucha microbiota can grow and multiply normally when stored in the range of 20–50 °C, and the formation of the tea bacterial film and the growth of bacterial and yeast colonies can still be observed after 3 days of storage at 50 °C. However, when the temperature exceeds the heat tolerance threshold of 60 °C, the biosynthesis of the microbial film is completely inhibited. Therefore, appropriate fermentation and storage temperatures can enhance microbial activity and enzymatic reactions, leading to an improvement in the quality of kombucha. For inactivated kombucha microbiota, which are marketed as beverages, it is usually necessary to store them at room temperature to ensure their stability and avoid a further fermentation process [13].

## 5. Microbial Composition and Synergistic Metabolic Mechanisms in Kombucha Fermentation

Kombucha fermentation relies on the synergistic action of microorganisms, such as *Saccharomyces*, Acetobacter, and Lactobacillus, and the composition and proportion of these microbial communities significantly influence the quality of the product. There are differences in the microbial composition of kombucha starter cultures (SCOBY) from different sources, which results in variations in the fermentation process and the quality of the final product [7]. *Saccharomyces*, during the early stages of fermentation, hydrolyze sucrose into glucose and fructose and produce ethanol. Their activity and species influence alcohol production and sugar metabolism. *Acetobacter* are responsible for oxidizing ethanol to acetic acid and synthesizing organic acids such as glucuronic acid, and their quantity and species affect the yield and types of organic acids produced. The interactions among microbial communities are also crucial. For example, ethanol produced by *Saccharomyces* serves as a substrate for Acetobacter, while the acidic environment produced by Acetobacter influences yeast growth and metabolism. An imbalance in microbial composition may lead to fermentation failure or a decline in product quality, resulting in issues such as off-flavors or spoilage.

### 5.1. Microbial Composition of Kombucha and Its Synergistic Fermentation Mechanism

The fermentation system of kombucha consists of the sequential metabolic synergism between *Saccharomyces, Acetobacter*, and *Lactobacillus*. This unique microbial composition and metabolic mechanism play a crucial role during the fermentation process of kombucha. As shown in Figure 3, *Saccharomyces*, *Acetobacter*, and *Lactobacillus* form a dynamic symbiotic network through a “sugar metabolism–ethanol oxidation–organic acid accumulation” cascade reaction. The interactions between their metabolic products and environmental factors collectively drive the progression of the fermentation process [14].

#### 5.1.1. Fermentation Initiation Phase: Yeast-Dominated Sugar Alcohol Conversion

During the initial phase of fermentation (0–72 h), *Saccharomyces*, as the predominant microbial group, play a crucial role in initiating fermentation-related metabolic processes. Several species of the genus *Saccharomyces* have been identified in kombucha, such as *Zygosaccharomyces* and *Brettanomyces*, which exhibit genetic diversity [15]. These facultative anaerobic *Saccharomyces* play a role through a two-stage metabolic process, as follows: first, by secreting sucrase (EC 3.2.1.26) to hydrolyze sucrose into glucose and fructose, and then converting the monosaccharides into ethanol and CO_2_ through the Embden–Meyerhof–Parnas (EMP) pathway [16,17].

#### 5.1.2. Mid-Fermentation Phase: Ethanol–Acetic Acid Reaction Mediated by Acetobacter

During the ethanol accumulation phase (72–168 h), the metabolic activity of *Acetobacter* significantly increases. Strains of Acetobacter, such as *Komagataeibacter xylinus*, play a dominant role in this phase. *Acetobacter* gradually oxidize ethanol to acetic acid through membrane-bound alcohol dehydrogenase (ADH, EC 1.1.1.1) and aldehyde dehydrogenase (ALDH, EC 1.2.1.3) [18]. Additionally, *Acetobacter* directly oxidize glucose to gluconic acid via membrane-bound glucose dehydrogenase (GDH, EC 1.1.5.2) [19]. During this period, the *Acetobacter* secrete β-1,4-glucan chains to form a cellulose membrane (SCOBY), with the outer layer of the membrane being rich in oxygen, facilitating aerobic metabolism of *Acetobacter*, while the lower layer of the membrane provides a low-oxygen environment, supporting the maintenance of *Saccharomyces* activity [20].

#### 5.1.3. Late Fermentation Phase: *Lactic Acid Bacteria*’s Auxiliary Acidification

In the final phase of fermentation (>168 h), *Lactobacillus* (e.g., *Lactobacillus plantarum*) enhance the stability of the fermentation system through the following mechanisms. *Lactobacillus* utilize the 6-phosphogluconate pathway for heterofermentative lactic acid fermentation, producing L-lactic acid and trace amounts of D-lactic acid [21]. Additionally, *Lactobacillus* synthesize terpenoid compounds such as diacetyl (buttery aroma) and ethyl phenylacetate (nutty aroma), which synergistically interact with *Acetobacter*’s metabolic products to generate aromatic effects. The production of lactic acid increases the total acidity, and under conditions of a pH below 4.0 and temperatures below 30 °C, it helps extend the shelf life of kombucha [22]. The production of metabolites by *lactic acid bacteria* helps to enhance the growth of *Saccharomyces* and Acetobacter, especially in low-oxygen environments, where the metabolites of *lactic acid bacteria* can serve as a carbon source for the yeasts to further promote the fermentation process [23].

During the fermentation of kombucha microbiota, *Saccharomyces, Acetobacter,* and *Lactobacillus* form a metabolic cascade reaction through the flow of substances and energy. First, yeast converts sugar to ethanol, *acetic acid bacteria* further convert ethanol to acetic acid, and *lactic acid bacteria* convert acetic acid to lactic acid, forming a continuous metabolic chain. In addition, the cross-feeding effect of Lactobacillus with *Saccharomyces* and Acetobacter significantly enhances the stability of the overall fermentation process, especially Lactobacillus is able to produce lactic acid by metabolizing acetic acid, which, in turn, regulates the acidity of the fermentation environment and promotes the growth of other microbial communities [22].

### 5.2. Bioactive Compounds in Kombucha

#### 5.2.1. Phenolic Compounds

The phenolic compounds in kombucha primarily include tea polyphenols, catechins, and theaflavins, among others [24]. Among these, catechins exhibit high instability during fermentation, with their concentration and stability closely related to fermentation time, tea variety, and fermentation conditions [25]. For instance, the catechin content in green tea is typically higher, often exceeding that of oolong tea, white tea, and black tea [26]. Furthermore, the antioxidant activity of kombucha is mainly attributed to phenolic compounds, which can effectively scavenge DPPH radicals, ABTS radicals, and inhibit linoleic acid peroxidation [9]. Related studies have shown that the total polyphenol and flavonoid contents in kombucha increase during fermentation. This increase is due to the enhanced activity of conversion enzymes during fermentation, leading to the accumulation of phenolic compounds and thereby enhancing antioxidant activity [27]. This antioxidant activity is closely related to the content and molecular structure of phenolic compounds in kombucha, where multiple hydroxyl groups in catechins contribute to their strong antioxidant capacity.

#### 5.2.2. Organic Acids

The organic acids produced during kombucha fermentation are the primary contributors to its characteristic sour taste and preservative function. Studies have shown that the organic acid composition in kombucha exhibits significant metabolic diversity [10]. The sour taste profile is primarily determined by acetic acid, glucuronic acid, and their derivatives. Acetic acid, as the major organic acid, contributes a sharp, pungent sour taste, while glucuronic acid imparts a relatively softer and refreshing acidic sensation [28]. The dynamic accumulation of these acidic compounds is closely related to the synergistic metabolism of the microbial community: *Saccharomyces* preferentially utilize fructose through the glycolytic pathway to produce ethanol, while *Acetobacter* oxidize glucose at the C-6 position to generate the mild acid taste of glucuronic acid, simultaneously converting ethanol into acetic acid [24]. In addition to these primary organic acids, kombucha also contains other organic acids, such as citric acid, malic acid, tartaric acid, and succinic acid [10]. The presence of these organic acids increases the acidity of kombucha, thereby inhibiting the growth of undesirable microorganisms.

#### 5.2.3. Other Components

Kombucha also contains other bioactive components, such as vitamins, amino acids, and bacteriocins. Although present in smaller quantities, these components play significant roles. The vitamins in kombucha are mainly B vitamins (B1, B2, B6, and B12) and vitamin C [14]. These vitamins are essential elements for the human body and offer health benefits, such as antioxidant properties, scurvy prevention, and immune support. The *Lactobacillus* in kombucha metabolize and produce bacteriocins, which are antimicrobial peptides that effectively inhibit the proliferation of pathogenic and spoilage microorganisms. In this way, they, along with organic acids, contribute to the creation of a multi-layered antimicrobial barrier [29].

## 6. Health Benefits of Kombucha

The health effects of kombucha result from the synergistic action of its bioactive compounds, including phenolic compounds, organic acids, sugars, and vitamins (Table 1). Existing research suggests that these components participate in physiological regulation through a multi-target mechanism, primarily manifesting in the following aspects [4,30,31].

### 6.1. Antioxidant Activity

The antioxidant properties of kombucha are primarily attributed to the synergistic effect between phenolic derivatives produced during microbial metabolism in fermentation and the matrix components of the tea leaves [7,37]. Kaewkod et al. systematically revealed the impact of different tea matrices on the antioxidant activity of the final products [32], finding that the DPPH radical scavenging activity of green tea was significantly higher than that of kombucha products derived from black tea and oolong tea.

In the symbiotic system of kombucha, *Saccharomyces* and *Acetobacter* secrete extracellular enzymes (such as polyphenol oxidase and glycoside hydrolases) to catalyze the biotransformation of substrates. This process promotes the enzymatic hydrolysis of macromolecular polyphenol complexes, generating small-molecule phenolic acids and flavonoid aglycones with higher bioactivity [33]. At the same time, microbial metabolism effectively releases bound bioactive substances from the matrix, thereby increasing the concentration of phenolic compounds and other active components [33,38]. Therefore, the microorganisms in kombucha enhance the bioavailability and antioxidant properties of active compounds, thereby improving its nutritional benefits and functional value as a health-promoting beverage [38].

### 6.2. Antimicrobial Properties

Kombucha exhibits significant antimicrobial effects against Gram-positive bacteria (e.g., *Staphylococcus aureus* and *Listeria monocytogenes*) and Gram-negative bacteria (e.g., *Escherichia coli* and *Pseudomonas aeruginosa*) [39,40]. Its antimicrobial mechanisms are multifaceted, as follows: (1) organic acids in the fermentation products, primarily acetic acid, lower the environmental pH through protonation, disrupting the integrity of the pathogen cell membrane [34,41]; (2) phenolic compounds interfere with the microbial enzyme systems [42]; and (3) bacteriocin-like substances produced during metabolism directly inhibit the proliferation of pathogens [35].

### 6.3. Anti-Inflammatory and Immune-Modulatory Activity

Kombucha can intervene in pathogen colonization and exert anti-inflammatory effects by downregulating the expression levels of pro-inflammatory cytokines (such as TNF-a and IL-6), thereby alleviating inflammation [43]. Wang found that kombucha exhibited significant anti-inflammatory effects in a mouse sepsis model induced by lipopolysaccharide (LPS) [44]. Kombucha downregulated the expression levels of pro-inflammatory cytokines (such as TNF-α and IL-6), alleviated the inflammatory response, mitigated tissue pathological damage, and inhibited LPS-induced NF-κB transcription factor signaling in septic mice, demonstrating potent anti-inflammatory activity [44]. Additionally, another study discovered that kombucha fermented with goji berries and honeysuckle improved immune function in BALB/c mice [45]. These findings suggest that kombucha and its derivatives have significant anti-inflammatory and immune-modulating effects, highlighting their potential for application.

### 6.4. Anti-Cancer Properties

Some bioactive components in kombucha confer anti-cancer properties. Kombucha exhibits anti-cancer effects through multiple mechanisms, including the following: (1) Inhibition of cancer cell proliferation. For instance, kombucha fermented with winter tea demonstrated inhibitory effects on cervical cancer (HeLa), colon cancer (HT-29), and breast cancer (MCF-7) cells [46]. (2) Direct cytotoxicity and blockage of metastatic pathways. The ethyl acetate component in kombucha exhibits direct cytotoxic effects on kidney cancer (786-O) and osteosarcoma (U2OS) cells. Furthermore, kombucha reduces the invasiveness of lung cancer (A549) cells by inhibiting the activity of matrix metalloproteinases (MMP-2/MMP-9), thereby blocking cancer cell metastasis [14]. (3) Enhanced expression of antioxidant enzymes: tea polyphenols (e.g., catechins and epicatechins) in kombucha microbiota activate the Nrf2 pathway through multiple mechanisms, thereby inhibiting oxidative stress and inflammatory responses [47]. Nrf2 activation is commonly associated with enhanced expression of antioxidant enzymes, which has been implicated in cancer inhibition. However, most studies on the anti-cancer properties of kombucha rely on in vitro research, with a lack of clinical evidence to support these findings, thus highlighting several limitations.

### 6.5. Other Health Benefits

Preliminary studies suggest that kombucha may have potential benefits such as hepatoprotective effects [36], reducing gastric ulcers [48], and anti-diabetic properties [49]. However, the mechanisms underlying these effects still require further verification and research.

## 7. Limitations of the Study

This review provides a detailed literature synthesis of the fermentation process, microbial community synergistic metabolic mechanisms, and health benefits of kombucha. However, several limitations still exist. First, existing research primarily focuses on laboratory-scale studies, and there are significant differences in the fermentation conditions and microbial strains used across studies, which may limit the comparability and reproducibility of the results [10,50]. Second, although factors such as fermentation time, temperature, and strain type significantly affect the quality and health benefits of kombucha, most existing studies focus on the impact of single factors and lack a systematic analysis of the interactions among these factors [12,51,52]. Therefore, there is still a lack of comprehensive, multi-dimensional research on how to optimize fermentation conditions to enhance the health benefits of kombucha. Finally, although the health benefits of kombucha are widely recognized, most of the current research is based on in vitro experiments. Specifically, the mechanisms and effects in areas such as antioxidant, antimicrobial, and immune modulation require more clinical and long-term studies for validation and further analysis [7,32,53]. In addition, despite our detailed discussion of the metabolic cascade reactions during *Acetobacter* fermentation in the paper, specific enzymatic kinetic data (e.g., Km and Vmax values) on alcohol dehydrogenase (ADH) and aldehyde dehydrogenase (ALDH) in Acetobacter spp. are currently lacking. The lack of these data limits our comprehensive understanding of the mechanisms of enzyme catalysis in *acetic acid bacteria* during kombucha microbiota fermentation. Future studies could focus on the kinetic properties of these enzymes, especially in the fermentation process of Acetobacter, thus providing a theoretical basis for further optimization of the fermentation process of Camellia sinensis.

In conclusion, although kombucha, as a functional beverage, theoretically offers a wide range of health benefits, further systematic research and large-scale clinical validation are required to better understand the specific effects of its fermentation process, microbial community interactions, and health benefits under varying conditions.

## 8. Conclusions

Kombucha, a traditional fermented beverage with a long history, offers significant health benefits. Its primary bioactive components, phenolic compounds and organic acids, contribute to its antioxidant, antimicrobial, and immune-modulatory functions. This review systematically analyzes the synergistic metabolic mechanisms of the microbial community during kombucha fermentation and their impact on health benefits. *Saccharomyces, Acetobacter*, and *Lactobacillus* play key roles in the “sugar metabolism–ethanol oxidation–organic acid accumulation” metabolic chain. During fermentation, *Saccharomyces* convert sugars into ethanol, *Acetobacter* convert ethanol into acetic acid, and *Lactobacillus* regulate the overall acidity by producing lactic acid and other organic acids in the later stages. This metabolic network is significantly regulated by factors such as fermentation time, temperature, and strain selection. The interaction between metabolic products and fermentation conditions drives the fermentation process, ultimately affecting the quality and health benefits of kombucha. Therefore, optimizing fermentation conditions is crucial for enhancing the bioactive components of kombucha and improving its health benefits.

## Figures and Tables

**Figure 1 biology-14-00952-f001:**
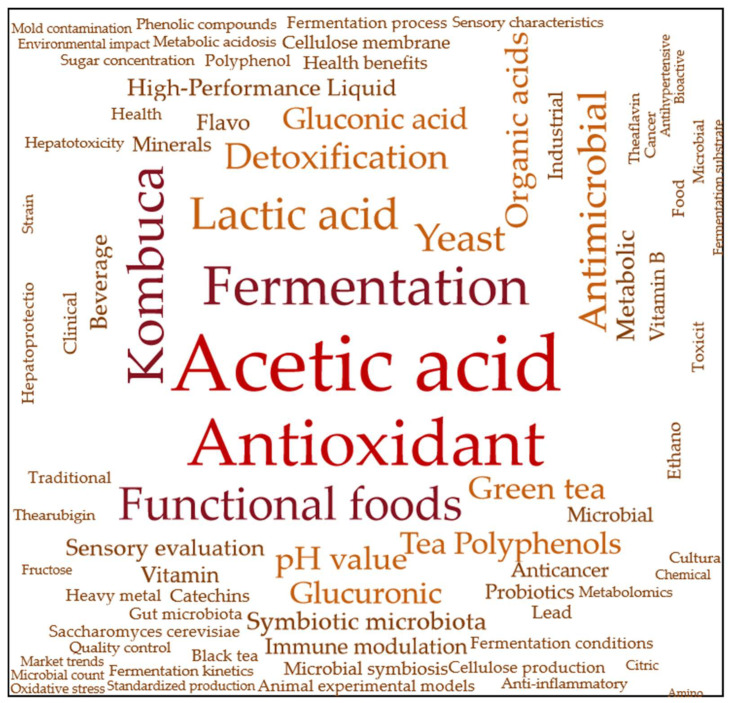
Word frequency analysis.

**Figure 2 biology-14-00952-f002:**
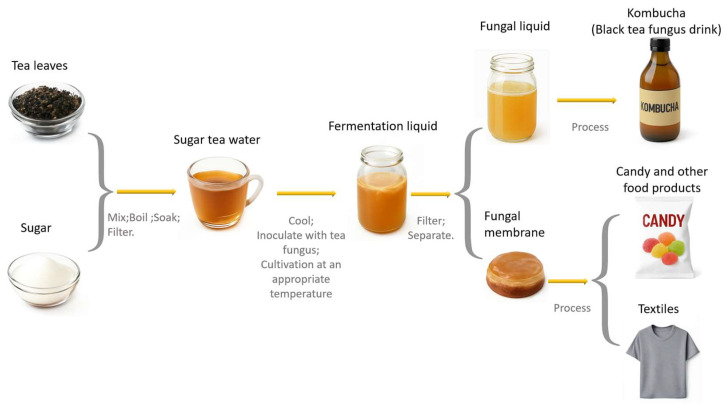
Kombucha process flow chart.

**Figure 3 biology-14-00952-f003:**
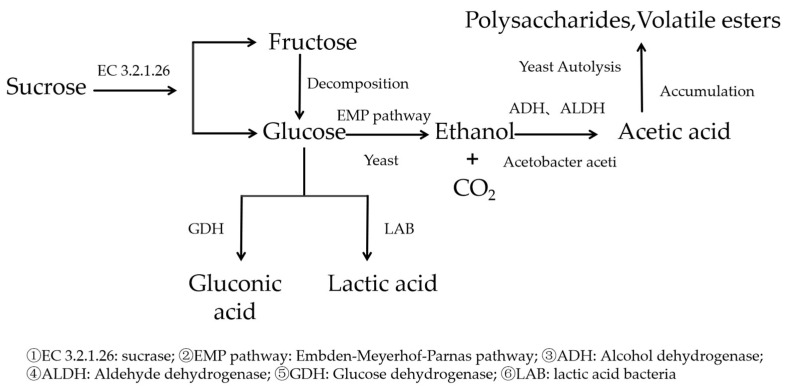
Synergistic metabolic mechanisms in kombucha fermentation.

**Table 1 biology-14-00952-t001:** Health effects of kombucha.

Health Effects	Active Ingredient	References
Antioxidant activity	Catechins	[32] Kaewkod T., 2019
	Flavonoid aglycogens, phenolic acids	[33] Bhattacharya S., 2013
Antimicrobial activity	acetic acid	[34] Valiyan F., 2021
	Catechins, thearubigins, theaflavins	[33] Bhattacharya S., 2013
	Gallic acid, caffeine	[33] Bhattacharya S., 2013
	Bacteriocins	[35] Atieh Darbandi, 2022
Anti-inflammatory and immunomodulatory	Vitamins	[7] Leal J.M., 2018
Anti-cancer activity	Lactic acid, ascorbic acid	[32] Kaewkod T., 2019
	Acetic acid, glucuronic acid	[32] Kaewkod T., 2019
Liver detoxification kinetic energy	Glucuronic acid	[36] Toda M., 2008

## Data Availability

Data are contained within the article.

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
