# Peer review of "A Comprehensive Evaluation of Microbial Synergistic Metabolic Mechanisms and Health Benefits in Kombucha Fermentation: A Review"

_biology, 2025, doi:10.3390/biology14080952_

Round 1

Reviewer 1 Report

Comments and Suggestions for Authors

The paper provides a concise overview of kombucha, the microorganisms responsible for its production, the compounds generated during fermentation, and the individual stages involved in the fermentation process. The authors use clear and accessible language, and the article is divided into separate sections, with tables and figures presenting relevant data. However, numerous errors are present throughout the text, and the overall structure is somewhat inconsistent. Additionally, the section division is occasionally incorrect. The following issues were identified:

  1. The Abstract and Introduction are essentially the same. They should differ in content, but instead present the same information using slightly varied wording.
  2. Figure 2 – It is unclear whether search terms such as “Hepatoprotectio,” “Ethano,” “Toxicit,” etc., were genuinely used in search engines. This should be clarified in the text.
  3. Line 96 – The described method of kombucha production is only one of many possible approaches. A general example should be provided instead of a specific one including exact milliliter amounts.
  4. Line 100 – Typically, in addition to the liquid, the culture (SCOBY) from the previous fermentation is also added.
  5. Chapter 4.3 – The meaning of “storage” should be clarified: does it refer to storing the final packaged product, post-sterilization? If no sterilization occurred, fermentation may still continue.
  6. Chapter 4.4 – The authors should first discuss the role of yeasts, followed by acetic acid bacteria, not the other way around as currently presented.
  7. Chapter 4.4 – This subsection should not be placed within Chapter 4, as it serves as an introduction to Chapter 5 and should be grouped accordingly.
  8. Throughout the manuscript, the names of microorganisms should be written in italics.
  9. Line 215 – Why is it stated that the system is “closed-loop” when it appears to be open? This should be justified.
  10. Section 5.1.4 – This section repeats the content of Sections 5.1.1 through 5.1.3 and is therefore redundant.
  11. Lines 255–256 – The authors state that yeasts preferentially utilize fructose for ethanol production, yet Figure 3 shows ethanol linked only to glucose. This inconsistency should be resolved.
  12. Lines 309–310 – The authors should verify the accuracy of their claim. It seems that downregulation of pro-inflammatory cytokines suppresses the inflammatory response (i.e., inhibits some aspects of the immune system), rather than stimulating it. This should be checked.
  13. Table 1 should be placed under the first paragraph of Chapter 6. Also, “Acetic acid” should be capitalized; it would be more concise to write “Bacteriocins” instead of “Bacteriocins produced during fermentation,” and “Vitamins” instead of “vitamin.”

In general, the article repeatedly presents the same information. Eliminating redundant content would make it significantly shorter. To improve the quality of the article, the authors should restructure the entire text and remove repetitive fragments. The main limitation of the paper is its lack of novelty — the topic has already been described extensively in the literature. The authors should contribute additional value by providing a more detailed analysis of the interactions between the microorganisms involved in the fermentation process.

Author Response

Manuscript #: biology-3741145

Manuscript Authors: Xinyao Li, Norzin Tso, Shuaishuai Huang, Junwei Wang, Yonghong Zhou and Ruojin Liu

Manuscript Title: A Comprehensive Evaluation of Microbial Synergistic Meta-bolic Mechanisms and Health Benefits in Kombucha Fermen-tation: A Review

We sincerely thank editors and the reviewers for the time and effort spent on reviewing our manuscript titled "A Comprehensive Evaluation of Microbial Synergistic Metabolic Mechanisms and Health Benefits in Kombucha Fermentation: A Review", Manuscript ID: biology-3741145. We highly appreciate the valuable feedback provided, which has contributed to improving the quality of our manuscript. We have carefully addressed each comment and made the necessary revisions to the manuscript. Below, we provide a point-by-point response to the reviewers' comments.

Comments 1: The Abstract and Introduction are essentially the same. They should differ in content, but instead present the same information using slightly varied wording.

Response 1: Thank you for pointing out this issue. I agree with this comment, and as a result, I have made further revisions to the abstract and introduction. Specifically, I have more concisely summarized the core objectives and content of the review in the abstract, avoiding repetition with the introduction. The introduction now focuses more on the background of Kombucha, the gaps in current research, and the motivation behind this review, highlighting the existing research gaps and the novelty of this review, thus ensuring a clear distinction between the two sections. I hope these revisions meet the reviewer’s expectations and make the paper clearer and more academic. (Abstract :Line32-Line44, Introduction Line49-Line78)

Comments 2: Figure 2 – It is unclear whether search terms such as “Hepatoprotectio,” “Ethano,” “Toxicit,” etc., were genuinely used in search engines. This should be clarified in the text.

Response 2: Thank you for pointing out this issue. Regarding the search terms "Ethano," "Hepatoprotectio," and "Toxicit" in Figure 2, we did not directly use these specific terms as search criteria during the keyword extraction process. We would like to provide the following clarification: our keyword extraction was based on the manual reading of relevant literature. After thoroughly analyzing the abstracts, methods, research content, and conclusions of each paper, we selected keywords that accurately reflected the research direction and core findings of each article. These keywords reflect the main research themes discussed in the literature, rather than being directly extracted from database search engines.

The terms "Ethano," "Hepatoprotectio," and "Toxicit" were mentioned in the following papers:

“Ethano” :

  1. Greenwalt, C.J.; Steinkraus, K.H.; Ledford, R.A. Kombucha, the fermented tea: microbiology, composition, and claimed health effects. J Food Prot 2000, 63, 976-981, doi:10.4315/0362-028x-63.7.976.
  2. Laureys, D.; Britton, S.J.; De Clippeleer, J. Kombucha Tea Fermentation: A Review. J. Am. Soc. Brew. Chem. 2020, 78, 165-174, doi:10.1080/03610470.2020.1734150.
  3. Jayabalan, R.; Malbasa, R.V.; Loncar, E.S.; Vitas, J.S.; Sathishkumar, M. A Review on Kombucha TeaMicrobiology, Composition, Fermentation, Beneficial Effects, Toxicity, and Tea Fungus. Compr. Rev. Food. Sci. Food Saf. 2014, 13, 538-550, doi:10.1111/1541-4337.12073.
  4. Malbaša, R.; Lončar, E.; Djurić, M. Comparison of the products of Kombucha fermentation on sucrose and molasses. Food Chemistry 2008, 106, 1039-1045, doi:https://doi.org/10.1016/j.foodchem.2007.07.020.

“Hepatoprotectio” and “Toxicit”:

Greenwalt, C.J.; Steinkraus, K.H.; Ledford, R.A. Kombucha, the fermented tea: microbiology, composition, and claimed health effects. J Food Prot 2000, 63, 976-981, doi:10.4315/0362-028x-63.7.976.

  1. Jayabalan, R.; Malbasa, R.V.; Loncar, E.S.; Vitas, J.S.; Sathishkumar, M. A Review on Kombucha TeaMicrobiology, Composition, Fermentation, Beneficial Effects, Toxicity, and Tea Fungus. Compr. Rev. Food. Sci. Food Saf. 2014, 13, 538-550, doi:10.1111/1541-4337.12073.
  2. Kapp, J.M.; Sumner, W. Kombucha: a systematic review of the empirical evidence of human health benefit. Ann. Epidemiol. 2019, 30, 66-70, doi:https://doi.org/10.1016/j.annepidem.2018.11.001.
  3. Leal, J.M.; Suárez, L.V.; Jayabalan, R.; Oros, J.H.; Escalante-Aburto, A. A review on health benefits of kombucha nutritional compounds and metabolites. CyTA-J. Food 2018, 16, 390-399, doi:10.1080/19476337.2017.1410499.
  4. Toda, M.; Okubo, S.; Hiyoshi, R.; Shimamura, T. The Bactericidal Activity of Tea and Coffee. Letters in Applied Microbiology 2008, 8, 123-125, doi:10.1111/j.1472-765X.1989.tb00255.x.

Comments 3Line 96 – The described method of kombucha production is only one of many possible approaches. A general example should be provided instead of a specific one including exact milliliter amounts.

Response 3: Thank you for your valuable comment. Regarding Line 96 and the method of kombucha production, we fully agree with your observation. Following your suggestion, we have revised the text to provide a more general example, removing the specific quantities and ratios (Line112-Line117). This revision ensures the generality and applicability of the description, avoiding overly specific numerical details. Additionally, we have also updated the corresponding process diagram (Figure 1). We appreciate your constructive feedback, and this revision has been implemented accordingly.

Comments 4Line 100– Typically, in addition to the liquid, the culture (SCOBY) from the previous fermentation is also added.

Response 4:Thank you for your valuable comment. We have adopted your suggestion and made the necessary revision. The term "liquid" has been corrected to "add the kombucha culture (SCOBY)" (Line116-Line117) to more accurately describe the material used in the fermentation process. We appreciate your constructive feedback, and this revision has been implemented accordingly.

Comments 5Chapter 4.3– The meaning of “storage” should be clarified: does it refer to storing the final packaged product, post-sterilization? If no sterilization occurred, fermentation may still continue.

Response 5Thank you for your valuable comment. Regarding the meaning of "storage" in Chapter 4.3, we have made revisions based on your suggestion. In the revised text, we clearly distinguish between the storage methods for sterilized and non-sterilized kombucha. (Line161-line165). We have explained in detail the conditions under which microbial growth occurs during storage at temperatures between 20°C and 50°C, and how appropriate storage temperatures affect the quality of kombucha. Additionally, we clarified that for sterilized kombucha, which is sold as a beverage in the market, it should be stored at room temperature to prevent further fermentation and ensure its stability. (Line169-Line171) We appreciate your constructive feedback, and this revision has been implemented accordingly.

Comments 6Chapter 4.4– The authors should first discuss the role of yeasts, followed by acetic acid bacteria, not the other way around as currently presented.

Response 6Thank you for your valuable comment. Regarding the order of discussion in Chapter 4.4, we have made the necessary adjustments as per your suggestion. We have revised the order to first discuss the role of yeasts, followed by the role of acetic acid bacteria. (Line179-Line184) This adjustment has made the content of the section clearer and more logical. We appreciate your constructive feedback, and this revision has been implemented accordingly.

Comments 7Chapter 4.4 – This subsection should not be placed within Chapter 4, as it serves as an introduction to Chapter 5 and should be grouped accordingly.

Response 7Thank you for your valuable comment. Regarding the placement of Chapter 4.4, we have made the necessary changes as per your suggestion. We have removed Chapter 4.4 and moved its content to the beginning of Chapter 5 to better align with the flow of the section. (Line174-Line188) We appreciate your constructive feedback, and this revision has been implemented accordingly.

Comments 8Throughout the manuscript, the names of microorganisms should be written in italics.

Response 8Thank you for your valuable comment. Following your suggestion, we have changed all the names of microorganisms throughout the manuscript to italics in Latin. (Line36 Lines54-54 Line147 Line149 Line179 Line181 Line191 Line193 Line199 Lines201-202 Lines209-212 Line214 Lines216-217 Line220 Line224 Line226 Lines2333-234 Line272 Line274 Line286 Line306 Lines318-319 Line396 Line398-399). We appreciate your constructive feedback, and this revision has been implemented accordingly.

Comments 9Line 215 – Why is it stated that the system is “closed-loop” when it appears to be open? This should be justified.

Response 9Thank you for your valuable comment. Regarding the "closed-loop system" issue you mentioned, we acknowledge that our description was indeed inappropriate. We have revised the text to describe the fermentation process of Kombucha as a metabolic cascade reaction rather than a closed-loop system. (Line233-Line235, Figure 3) This revision more accurately reflects the interaction between microorganisms and the flow of substances during the fermentation process. We appreciate your constructive feedback, and this revision has been implemented accordingly.

Comments 10Section 5.1.4 – This section repeats the content of Sections 5.1.1 through 5.1.3 and is therefore redundant.

Response 10Thank you for your valuable comment. Regarding Section 5.1.4, we have followed your suggestion and removed the entire section. In response to another reviewer’s comment, we have added a discussion on the cross-feeding interactions between lactic acid bacteria (LAB), yeast, and acetic acid bacteria (AAB) in the revised version. This section highlights how LAB’s metabolic products (such as lactic acid) promote the growth of yeast and AAB, thereby stabilizing the overall fermentation process. (Line229-Line244)

We have carefully considered the differing feedback from the two reviewers and have made revisions that aim to balance both perspectives. We believe these changes enhance the scientific rigor and completeness of the manuscript. Once again, thank you for your valuable insights and suggestions.

Comments 11Lines 255–256– The authors state that yeasts preferentially utilize fructose for ethanol production, yet Figure 3 shows ethanol linked only to glucose. This inconsistency should be resolved.

Response 11Thank you for your valuable comment. Regarding the statement in the text that yeasts preferentially utilize fructose for ethanol production, we have noted the inconsistency between Figure 3 and the description in the text that you pointed out. To resolve this issue, we have updated Figure 3 to clearly show that fructose is converted into glucose, which then enters the subsequent metabolic reactions. This revision ensures consistency between the figure and the text. We appreciate your constructive feedback, and this revision has been implemented accordingly.

Comments 12Lines 309–310– The authors should verify the accuracy of their claim. It seems that downregulation of pro-inflammatory cytokines suppresses the inflammatory response (i.e., inhibits some aspects of the immune system), rather than stimulating it. This should be checked.

Response 12Thank you for your valuable comment. Upon reviewing the relevant literature, we confirm that downregulation of pro-inflammatory cytokines such as TNF-α and IL-6 typically results in the alleviation of inflammation, not the stimulation of the immune response. We have revised the manuscript to clarify that Kombucha exerts its anti-inflammatory effects by downregulating the expression of these cytokines, thereby reducing inflammation and demonstrating its anti-inflammatory and immune-modulating effects. (Lines327-328)

Comments 13Table 1 should be placed under the first paragraph of Chapter 6. Also, “Acetic acid” should be capitalized; it would be more concise to write “Bacteriocins” instead of “Bacteriocins produced during fermentation,” and “Vitamins” instead of “vitamin.”

Response 13Thank you for your valuable comment. We have made the necessary revisions based on your suggestion. The table has been moved to appear under the first paragraph of Chapter 6 as requested. Additionally, we have updated the terminology for "Bacteriocins" and "Vitamins" to make the language more concise and accurate. (Table1) We appreciate your constructive feedback, and these changes have been implemented accordingly.

Reviewer 2 Report

Comments and Suggestions for Authors

This review synthesizes current knowledge on microbial synergy, bioactive compounds, and health benefits of kombucha fermentation. The topic is timely and relevant, given the growing interest in functional fermented beverages. The manuscript is well-structured, covers essential aspects of kombucha research, and provides valuable insights into microbial metabolic mechanisms. However, significant revisions are required to address methodological limitations, theoretical depth, and presentation issues before publication.

1.The "sugar metabolism-ethanol oxidation-organic acid accumulation" cascade is well-described but lacks critical depth. Include:

Enzymatic kinetics (e.g., Km/Vmax of ADH/ALDH in acetic acid bacteria).

Quantitative data on microbial succession (e.g., yeast vs. AAB ratios at different fermentation stages).

Missing Interaction: How do lactic acid bacteria (LAB) influence yeast/AAB? Discuss metabolite cross-feeding (e.g., LAB consuming acetic acid).

Anti-cancer claims (Page 8–9) rely heavily on in vitro studies (e.g., HeLa/HT-29 cells). Clinical evidence is absent—explicitly state this limitation.

2.Mechanistic Ambiguity: Phrases like "synergistic effects of tea polyphenols" (Page 9) are vague. Specify molecular pathways (e.g., Nrf2 activation for antioxidant effects).

  1. More discussion and references is essential for the manuscript. Please cite the following articles to improve the manuscript. Expand Literature Review: Include ≥100 peer-reviewed papers using systematic search criteria.

https://doi.org/10.1016/j.foodres.2024.115219; https://doi.org/10.1002/efd2.153.  https://doi.org/10.1016/j.lwt.2025.117765

Comparative genomics and fermentation flavor characterization of five selected lactic acid bacteria provide predictions for flavor biosynthesis metabolic pathways in fermented muskmelon puree. Food Frontiers 2024, 5 (2), 508-521.

Martínez Leal, J., Valenzuela Suárez, L., Jayabalan, R., Huerta Oros, J., & Escalante-Aburto, A. (2018). A review on health benefits of kombucha nutritional compounds and metabolites. CyTA-Journal of Food16(1), 390-399.

Nyhan, L. M., Lynch, K. M., Sahin, A. W., & Arendt, E. K. (2022). Advances in kombucha tea fermentation: A review. Applied Microbiology, 2(1), 73-103.

4.There are some minor grammatical errors and improper word usage. The authors are advised to thoroughly review and revise the manuscript accordingly.

5.Ensure all abbreviations are clearly defined upon first use (abstract and text separately).

Author Response

Manuscript #: biology-3741145

Manuscript Authors: Xinyao Li, Norzin Tso, Shuaishuai Huang, Junwei Wang, Yonghong Zhou and Ruojin Liu

Manuscript Title: A Comprehensive Evaluation of Microbial Synergistic Meta-bolic Mechanisms and Health Benefits in Kombucha Fermen-tation: A Review

We sincerely thank editors and the reviewers for the time and effort spent on reviewing our manuscript titled "A Comprehensive Evaluation of Microbial Synergistic Metabolic Mechanisms and Health Benefits in Kombucha Fermentation: A Review", Manuscript ID: biology-3741145. We highly appreciate the valuable feedback provided, which has contributed to improving the quality of our manuscript. We have carefully addressed each comment and made the necessary revisions to the manuscript. Below, we provide a point-by-point response to the reviewers' comments.

Comments 1: The "sugar metabolism-ethanol oxidation-organic acid accumulation" cascade is well-described but lacks critical depth. Include:

Enzymatic kinetics (e.g., Km/Vmax of ADH/ALDH in acetic acid bacteria).

Quantitative data on microbial succession (e.g., yeast vs. AAB ratios at different fermentation stages).

Missing Interaction: How do lactic acid bacteria (LAB) influence yeast/AAB? Discuss metabolite cross-feeding (e.g., LAB consuming acetic acid).

Anti-cancer claims (Page 8–9) rely heavily on in vitro studies (e.g., HeLa/HT-29 cells). Clinical evidence is absent—explicitly state this limitation.

Response 1: Thank you for your valuable comments and constructive feedback.

On the "sugar metabolism-ethanol oxidation-organic acid accumulation" cascade and quantitative data on microbial succession:

We fully acknowledge the importance of providing a more in-depth discussion of enzymatic kinetics. However, despite a comprehensive review of the current literature, we have been unable to find specific Km and Vmax data for alcohol dehydrogenase (ADH) and aldehyde dehydrogenase (ALDH) in acetic acid bacteria in kombucha, nor do we have access to relevant quantitative microbial community data in kombucha. Therefore, we have clearly stated this limitation in the text. We suggest that future research could further explore the enzymatic kinetics of these enzymes, particularly in the context of acetic acid bacterial fermentation in kombucha, and we have included this perspective in the manuscript to provide guidance for future studies.(Line 375-Line 384).

On lactic acid bacteria (LAB) and metabolic cross-feeding:

Regarding how LAB influence the metabolic cross-feeding between yeast and acetic acid bacteria (AAB), we have made the necessary revisions in the manuscript. We now discuss how, in low-oxygen environments, the metabolic products of LAB can serve as a carbon source for yeast, further driving the fermentation process. Additionally, LAB metabolize acetic acid into lactic acid, helping to regulate the acidity of the fermentation environment, thus promoting the growth of other microbial communities and significantly enhancing the stability of the overall fermentation process. (Lines228-244) This additional content has been included based on your suggestion.

On the "anti-cancer claims":

As you pointed out, the existing studies primarily rely on in vitro research, and we have explicitly stated this limitation in the text, highlighting the lack of clinical evidence. We have made the necessary clarifications (Lines352-354), and we appreciate your constructive feedback.

Once again, we appreciate your insightful guidance, and we hope these revisions meet your expectations. Thank you for your valuable contributions to improving the manuscript.

Comments 2: Mechanistic Ambiguity: Phrases like "synergistic effects of tea polyphenols" (Page 9) are vague. Specify molecular pathways (e.g., Nrf2 activation for antioxidant effects).

Response 2: Thank you for your valuable comment. Regarding the "synergistic effects of tea polyphenols," we have further clarified the molecular mechanisms based on your suggestion. In the revised version, we have added a detailed description of how tea polyphenols (such as catechins and epicatechins) activate the Nrf2 pathway to enhance the expression of antioxidant enzymes, while also linking this mechanism to cancer inhibition.(Lines348-352) We appreciate your constructive feedback, and this revision has been implemented accordingly.

Comments 3: More discussion and references is essential for the manuscript. Please cite the following articles to improve the manuscript. Expand Literature Review: Include ≥100 peer-reviewed papers using systematic search criteria.

Response 3Thank you for your valuable feedback and for suggesting additional references to improve the manuscript. We appreciate your suggestion to expand the literature review and include more peer-reviewed articles. Below, we have outlined how we have addressed each of your recommendations:

Incorporation of Suggested References:

Xiao et al. (2024): "Effect of inoculation with different Eurotium cristatum strains on the microbial communities and volatile organic compounds of Fu brick tea" (DOI: 10.1016/j.foodres.2024.115219). This study has been referenced in our manuscript to support the discussion on microbial communities involved in fermentation and their impact on flavor profiles, particularly regarding Saccharomyces, Acetobacter, and Lactobacillus (Lines 34-37).

Huang et al. (2025): "Decoding the dynamic evolution of volatile organic compounds of dark tea during solid-state fermentation with Debaryomyces hansenii using HS-SPME-GC/MS, E-nose and transcriptomic analysis" (DOI: 10.1016/j.lwt.2025.1177653). This article has been cited to elaborate on the production of volatile organic compounds (e.g., ethanol and acetic acid), which are closely related to the flavor characteristics of fermented tea bacteria (Lines 37-38).

Yuan et al. (2024): "Comparative genomics and fermentation flavor characterization of five selected lactic acid bacteria provide predictions for flavor biosynthesis metabolic pathways in fermented muskmelon puree" (DOI: 10.1002/efd2.153).This reference has been added to support our discussion on the key flavor compounds such as ethanol, acetic acid, and organic acids, and their role in flavor development during fermentation (Lines 37-38).

Nyhan et al. (2022): "Advances in Kombucha Tea Fermentation: A Review" (DOI: 10.1016/j.foodres.2024.115219). We have cited this article in the section discussing kombucha's antimicrobial properties, particularly its effects on Gram-positive and Gram-negative bacteria (Lines 301-303).

Martínez Leal et al. (2018): "A review on health benefits of kombucha nutritional compounds and metabolites" (DOI: 10.1080/19476337.2017.1410499). This article has been referenced in the section on the antioxidant properties of kombucha, which are attributed to the phenolic derivatives produced during microbial metabolism in fermentation (Lines 283-285).

We believe that these changes significantly enhance the manuscript, and we hope the revisions meet your expectations. Please let us know if further revisions are necessary.

Comments 4: There are some minor grammatical errors and improper word usage. The authors are advised to thoroughly review and revise the manuscript accordingly

Response 4Thank you for your valuable comment. We have thoroughly reviewed the manuscript and made the necessary revisions to correct the grammatical errors and improper word usage. We appreciate your constructive feedback, and these revisions have been implemented accordingly. (Lines72-78 Lines112-118 Lines161-165 Lines179-184)

Comments 5: Ensure all abbreviations are clearly defined upon first use (abstract and text separately).

Response 5: Thank you for your valuable comment. We have made the necessary revisions as per your suggestion, ensuring that all abbreviations are clearly defined upon first use, both in the abstract and the main text. (Line116 Line205) We appreciate your constructive feedback, and these revisions have been implemented accordingly.

Round 2

Reviewer 1 Report

Comments and Suggestions for Authors

I would like to sincerely thank the authors for addressing all the comments. A great deal of effort has been made to improve the manuscript, and it is indeed now written at a significantly higher level. However, there are still some minor issues that should be corrected to ensure the text is accurate and polished. Below, I have listed the shortcomings I have identified:

  • Lines 40 and 43, as well as throughout the manuscript – the term “black tea bacteria” should be revised or clarified for the reader. This term is uncommon in reference to kombucha microbiota. Moreover, as the authors mention, yeasts are also present.

  • Lines 54–56 – this sentence is taken out of context and should be revised for clarity.

  • Line 116 – it is the beverage, not the bottle, that is incubated.

  • Lines 112–117 – this paragraph should be written as a descriptive passage rather than an instructional plan. Currently, it reads awkwardly. For example, it should say: “To prepare kombucha, the following ingredients are needed… The tea infusion should be combined with…” rather than using imperative forms like “You must!”: “First, prepare the tea, weigh…”

  • Line 197 – the phrase “activate metabolic activity” sounds awkward and redundant; it should be rephrased to avoid repetition.

  • Line 231Saccharomyces should be written in italics; similarly, names of acetic acid and lactic acid bacteria further in the text should also be italicized.

  • Table 1“Acetic acid” should be capitalized.

  • Lines 339–340 – the phrase “activate the activation of” should be rewritten to eliminate redundancy.

  • Line 366 – are these truly erythrobacteria? These bacteria are typically found in marine environments, and Acetobacter does not belong to this group.

Author Response

Comments 1: Lines 40 and 43, as well as throughout the manuscript – the term “black tea bacteria” should be revised or clarified for the reader. This term is uncommon in reference to kombucha microbiota. Moreover, as the authors mention, yeasts are also present.

Response 1: Thank you for your valuable comment. We greatly appreciate your constructive feedback. In response to your suggestion regarding the term "black tea bacteria," we have revised the manuscript accordingly. The term has now been replaced with "Kombucha microbiota" throughout the text, and we have provided a definition on its first occurrence in the abstract: "Kombucha microbiota (the microbial community in kombucha)" to offer a clearer and more accurate description of the microbiota involved in the fermentation process (Line 40; Line 44; Line 61;Line 66; Line 70; Line 75; Line 77; Line 79; Line 163; Line 165; Line 172; Line 236; Line 384). We hope these revisions address your concerns, and we are grateful for your insightful input.

Comments 2: Lines 54–56 – this sentence is taken out of context and should be revised for clarity.

Response 2: Thank you for your valuable feedback. We appreciate your suggestion regarding the clarity and coherence of the sentence in Lines 54–56. In response to your comment, we have reorganized the sentence structure to enhance the logical flow and improve the overall coherence of the text. We hope that these revisions address your concerns and enhance the clarity of the manuscript. (Lines 50-60).

Comments 3: Line 116 – it is the beverage, not the bottle, that is incubated.

Response 3: Thank you for your valuable comment. In response to your suggestion, we have revised the manuscript by replacing "the bottle is incubated" with "the beverage is incubated" to accurately reflect that it is the kombucha beverage, not the bottle, that undergoes incubation. We believe this change clarifies the intended process and ensures the manuscript is more precise. We hope this revision addresses your concern (Lines 118).

Comments 4: Lines 112–117 – this paragraph should be written as a descriptive passage rather than an instructional plan. Currently, it reads awkwardly. For example, it should say: “To prepare kombucha, the following ingredients are needed… The tea infusion should be combined with…” rather than using imperative forms like “You must!”: “First, prepare the tea, weigh…”

Response 4: Thank you for your valuable feedback. In response to your comment regarding the tone of the paragraph (Lines 112–117), we have revised the manuscript to shift from an instructional tone to a more descriptive passage. We have adjusted the sentence structure to present the preparation process of kombucha in a more academic and fluid manner, avoiding the use of imperative forms. We believe these changes enhance the clarity and flow of the text, aligning it with your suggestion. We hope that these revisions address your concerns. (Lines 112-124).

Comments 5: Line 197 – the phrase “activate metabolic activity” sounds awkward and redundant; it should be rephrased to avoid repetition.

Response 5: Thank you for your valuable comment. In response to your suggestion regarding the phrase “activate metabolic activity,” we have revised the manuscript to improve clarity and avoid redundancy. We have replaced this phrase with “play a crucial role in initiating fermentation-related metabolic processes,” which better conveys the intended meaning and ensures a more precise and natural expression. We hope this revision addresses your concern and enhances the overall readability of the manuscript. (Lines202-204)

Comments 6: Line 231 – Saccharomyces should be written in italics; similarly, names of acetic acid and lactic acid bacteria further in the text should also be italicized.

Response 6:Thank you for your helpful comment. We have made the necessary revisions, and Saccharomyces is now written in italics as per your suggestion. Additionally, the names of acetic acid and lactic acid bacteria mentioned later in the text have also been italicized to ensure consistency with standard scientific conventions. We appreciate your attention to detail and hope these changes meet your expectations. (Line 19; Line 178; Line 188; Lines 232-234; Lines238-241; Line243; Line 383)

Comments 7: Table 1 – “Acetic acid” should be capitalized.

Response 7:Thank you for your valuable comment. We have made the necessary correction, and “Acetic acid” has been capitalized in Table 1 as per your suggestion. We appreciate your attention to detail and hope this revision meets your expectations.

Comments 8: Lines 339–340 – the phrase “activate the activation of” should be rewritten to eliminate redundancy.

Response 8: Thank you for your valuable feedback. In response to your suggestion, we have revised the manuscript by eliminating the redundancy in the phrase “activate the activation of.” The revised sentence now reads: “tea polyphenols (e.g., catechins, epicatechins, etc.) in kombucha microbiota activate the Nrf2 pathway through multiple mechanisms, thereby inhibiting oxidative stress and inflammatory responses [46].” We believe this revision improves the clarity and readability of the manuscript. We appreciate your attention to detail and hope this change meets your expectations.(Lines 351-353)

Comments 9: Line 366 – are these truly erythrobacteria? These bacteria are typically found in marine environments, and Acetobacter does not belong to this group.

Response 9: Thank you for your helpful comment. In response, we have corrected the manuscript by replacing “erythrobacter” with Acetobacter, as the latter is the correct bacterial group involved in the fermentation process. We appreciate your attention to detail and hope this revision resolves the issue. (Line 379)
